# Substrate Optimization for Shiitake (*Lentinula edodes* (Berk.) Pegler) Mushroom Production in Ethiopia

**DOI:** 10.3390/jof9080811

**Published:** 2023-07-31

**Authors:** Buzayehu Desisa, Diriba Muleta, Tatek Dejene, Mulissa Jida, Abayneh Goshu, Pablo Martin-Pinto

**Affiliations:** 1Institute of Biotechnology, Addis Ababa University, Addis Ababa P.O. Box 1176, Ethiopia; buzayehudesisa@gmail.com (B.D.); dmuleta@gmail.com (D.M.); 2Ethiopian Forest Development, P.O. Box 24536, Addis Ababa 1000, Ethiopia; tatekdejene.bekele@uva.es; 3Sustainable Forest Management Research Institute, University of Valladolid, Avda. Madrid 44, 34071 Palencia, Spain; 4Bio and Emerging Technology Institute, Addis Ababa P.O. Box 5954, Ethiopia; mulaeabageda@gmail.com (M.J.); abayday2002@gmail.com (A.G.)

**Keywords:** sugarcane bagasse, substrate, yield, biological efficiency, shiitake

## Abstract

Edible mushrooms are seen as a way of increasing dietary diversity and achieving food security in Ethiopia. The aim of this study was to develop substrates using locally available agro-industrial by-products and animal manures to enhance the production of Shiitake (*Lentinula edodes*) mushrooms in Ethiopia. The hypothesis was *L. edodes* mushroom production on seven different substrates: 100% sugarcane bagasse (S1), 80% sugarcane bagasse, 20% cow dung (S2), horse manure (S3), chicken manure (S4), cottonseed hulls (S5), sugarcane filter cake (S6), and sugarcane trash (S7). Mushroom yield and biological efficiency were significantly affected by substrate type (*p* < 0.05). A significantly higher yield (434.33 g/500 g of substrate) and biological efficiency (86.83%) were obtained using substrate S4 while lower yield (120.33 g/500 g) and biological efficiency (24.33%) were obtained using substrate S7 than when using other substrates. The largest first flush of mushrooms was obtained on S4, and five flushes were produced on this substrate. S4 also had the highest biological efficiency, the highest nitrogen content, and the lowest C:N. Chicken manure is rich in nitrogen, magnesium, calcium, and potassium, which are crucial for Shiitake mushroom growth. Thus, substrate S4 would be a viable option for cultivating Shiitake mushrooms, particularly in regions where chicken manure is readily available. Substrate S2 also provided high yields and rapid fructification and would be a suitable alternative for Shiitake mushroom cultivation.

## 1. Introduction

Ethiopia has the second largest population in Africa [1] and is predominantly an agricultural society: approximately 85% of the population is involved in agricultural activities [2]. However, agricultural productivity in Ethiopia is declining, mainly due to climate change and unsustainable agricultural practices, leading to soil erosion and degradation, which ultimately affect crop productivity [2,3]. As a result, per capita food production is low, leading to food insecurity [4]. To address food security issues, there is a need to expand food options and make them more accessible [2]. Edible mushrooms have been identified as a way of increasing dietary diversity and achieving food security [5,6]. However, the availability and collection of wild edible mushrooms in Ethiopia are limited to the rainy season when they grow in abundance in forests and other natural habitats [7,8]. To increase the availability of mushrooms, there have been efforts in recent years to cultivate mushrooms under controlled environments [9,10]. However, mushrooms require specific substrates for their nutrition. Therefore, substrate formulation and optimization from available organic materials and agricultural wastes are crucial to the success of edible mushroom cultivation [11,12].

Shiitake (*Lentinula edodes* (Berk.) Pegler), the most commonly cultivated mushroom worldwide [13] together with *Agaricus bisporus*, can be grown all year round under controlled conditions [14] using available organic materials. As well as being rich in proteins, shiitake mushrooms are also a good source of dietary fiber, macro- and micronutrients, sugars, tocopherols, polyunsaturated fatty acids, and have low levels of saturated fatty acids [15]. Furthermore, shiitake mushrooms are well known for their medicinal properties [16] and have been shown to own various health benefits, including antitumor [17], antioxidant [18], antiviral, antibacterial, and cholesterol-lowering activities [17]. Therefore, shiitake mushroom cultivation could contribute significantly to food security by providing a nutritious, high-yielding, low-input, and environmentally sustainable source of food and income [19]. 

The type of mushroom and its cultivation can affect the choice and combination of substrate materials [20]. Shiitake produce lignocellulolytic enzymes and, hence, shiitake cultivation has traditionally involved the use of logs as a substrate [21]. However, different types of lignocellulosic material have also been shown to support the growth and fruiting of shiitake [20], including various agro-industrial residues. However, different substrates have different pH levels and contain different levels of nutrients, moisture, and other factors that can affect the growth and development of the mycelium and fruiting bodies and, therefore, can have a significant impact on mushroom quality and yield. For example, substrates derived from agricultural waste materials such as wheat straw, corn cobs, or sawdust provide a good source of carbon (C) and nitrogen (N) for the mycelium [22]. Other materials, such as poultry manure or coffee grounds, can provide additional nutrients and can help to increase mushroom yields [23]. Typically, an 80% hardwood sawdust and 20% additive mixture is used as the standard substrate formula for shiitake cultivation [23]. However, this study used a starch-based substrate (sugarcane bagasse) as the main substrate as used by [24,25] and supplemented it with nutrient additives such as cow dung, horse manure, chicken manure, or sugarcane filter cake, which we assumed would enhance mushroom yields due to their high nutrient content [26]. Mushrooms can grow and provide higher yields when using a substrate of optimal C:N [27]. Therefore, the hypothesis of this study would be there is a relationship between the C:N of the substrate and mushroom yield. 

Thus, the objective of this study was to investigate the effect of different substrate combinations comprising locally available agro-industrial by-products and animal manures on the growth and yield of shiitake mushrooms, including their effects on spawning time, days to pinhead formation, days to first harvest, yield, and biological efficiency.

## 2. Materials and Methods

### 2.1. Experimental Design

The experiment was carried out at the Forests Product Innovation Center in Addis Ababa, Ethiopia, between September and January 2022. The experiment was laid out in a completely randomized design with seven substrate treatments and three replications per treatment.

### 2.2. Culture Source and Spawn Preparation

A commercial strain of shiitake (*Lentinula edodes*) obtained from the Mycology Laboratory at Addis Ababa University College of Natural and Computational Science was used in this study. Shiitake mycelium was grown on potato dextrose agar medium, a typical formula in g/L of (PDA) (Eur.pharm.) with Potato peptone (4), Glucose (20) and Agar (15), in sterile Petri dishes under sterile conditions at 25 ± 2 °C in complete darkness for two weeks (Figure 1A). Polypropylene plastic bottles (500 mL) were filled three-quarters full with mother spawn substrate comprising 95% wheat grain, 4% gypsum and 1% calcium carbonate on a dry weight basis [28]. The spawn substrate was sterilized by autoclaving at 121 °C for 80 min [29]. After cooling down to room temperature, the sterilized substrate was inoculated with 10 g of actively growing shiitake mycelium and incubated at 25 ± 2 °C for 18 days in a fully dark room, by which time the grain was completely covered by mycelium (Figure 1B). Then after putting the cotton on the top of the bottle, it was sealed with tin tread-like ropes.

To prepare commercial bags of *L. edodes* spawn, 7.5 × 35.0 cm polypropylene bags were filled three-quarters full of tightly packed mother spawn substrate as described above and then sterilized by autoclaving at 121 °C for 80 min. After cooling down to room temperature, the sterilized grains were inoculated with 15 g of mother spawn on a *w*/*w* wet-weight basis, as described by Atila [22]. The inoculated bags were then maintained at 25 ± 2 °C for 12 days, by which time the grain was completely covered by mycelium (Figure 1C).

### 2.3. Substrate Preparation

Seven distinct agro-industrial by-products were selected to create supplement-based substrates based on the availability of locally sourced biomass and its sustainability for mushroom cultivation. The substrate treatments comprised 100% sugarcane bagasse (S1), 80% sugarcane bagasse, 20% cow dung (S2), horse manure (S3), chicken manure (S4), cotton seed hulls (S5), and sugarcane by-product—filter cake (S6) or trash (S7)— (Table 1). 

Sun-dried supplements and sugarcane bagasse were cut into smaller pieces of a specific length following the method described by Gaitán–Hernández et al. [30] and weighed separately. The bagasse and one of the six supplements were then mixed thoroughly by hand as per the substrate formulation shown in Table 1. Substrates were soaked in water for 24 h and then drained to reduce the moisture content to 60–65%. To each substrate, 1% calcium carbonate and 1% gypsum (on a dry weight basis) were added to adjust the pH and prevent clumping of the substrate, respectively.

Substrates (500 g wet wt) were placed in unused heat-resistant polypropylene bags (20 × 35 cm) and sterilized in an autoclave at 121 °C for 120 min. After sterilization, substrates were inoculated with 3% fresh *L. edodes* spawn, equivalent to 15 g for 500 g of substrates, in a laminar flow chamber and sealed before being transferred to a dark incubation room for the duration of the spawn running stage (temperature 25 °C; relative humidity 85 ± 5%). After 26–35 days, substrates were completely covered with mycelium and a dark-brown crust had developed. Bags were then exposed to daylight for 12 h per day at a temperature of 18 ± 2 °C and 80–90% relative humidity in a controlled room to induce fructification. Mushroom fruiting bodies were harvested from substrates when fruit bodies were mature, and gills were fully exposed. Mushrooms were picked with clean hands without harming the substrate. After each fructification of mushrooms had been harvested, the bags of substrate were re-soaked in water for three days in tap water to recover the moisture and then moved back to the fruiting room to facilitate pinhead formation and fructification. This process was repeated for three to five flushes. The mushroom fructification room and shelf arrangements were designed following Beje et al. [28]. 

### 2.4. Determination of L. edodes Cultivation Parameters 

Mushroom cultivation and fruiting were evaluated following the methods described by Iqbal et al. [31] using the following parameters: the time required for spawn running (d), the first appearance of pinhead formation (d), the first harvest (d), yield (in g), which was based on the total weight of three to five flushes (i.e., the weight of fresh mushrooms (g) harvested at maturity per 500 g of dry substrate, *w*/*w*), and biological efficiency (%), which was determined by dividing the fresh weight of harvested fruiting bodies (g) by the dry weight of the uninoculated substrate (g) × 100, following the method described by Atila [22]. Cap diameter cap (cm), stipe length (cm) and number of fruiting bodies were also recorded for each bag of substrate. We have used a sliding caliper for the mushroom’s physical measurements.

### 2.5. Substrate Analyses

Prior to inoculation, samples of each of the seven substrates were oven-dried at 60 °C for 48 h and then ground up and passed through a sieve with 1-mm^2^ mesh. Moisture and total ash content were determined using the Ethiopian Standards method ES1032-1:2005. The lignocellulosic content—that is, the alcohol-toluene solubility, the Klason lignin content, and the cellulose and hemicellulose content of substrates—were measured using the standard method of the American Society for Testing and Materials D 1107-56, direct extraction with aqueous alkali, and Kurchner-Hoffer methods, respectively. The total crude fiber was determined using BCTL/SOP/M017.01 in the Agricultural Food Product Analysis Manual, which is based on the International Organization for Standardization’s ISO 5498:1981 agricultural food products—general method for the determination of crude fiber—general method. The Soxhlet extraction technique was used to determine the crude fat and crude protein content of substrates [32]. The N content of substrates was measured using the Kjeldahl method [33]. The C content was calculated by determining the fixed C content, volatile matter content, and ash content of the biomass, as described by Dai et al. [34]. The C:N ratio was calculated as C/N. Macro- (K, Ca, Mg, and Na) and microelement (Fe and Zn) concentrations were quantified by performing Microwave Plasma Atomic Emission Spectroscopy.

### 2.6. Statistical Analysis

Substrates were compared based on their chemical composition and mineral content. In addition, the impact of the substrate on the growth and yield of shiitake mushrooms based on spawning time, days to pinhead formation, days to first harvest, yield, and biological efficiency were assessed. Data analyses were performed using Statistical Package for Social Sciences (SPSS) version 20 [35]. Data were log-transformed when needed to achieve the parametric criteria of normality and homoscedasticity necessary for the analysis of variance. Differences between substrate options for the different variables were evaluated using a one-way analysis of variance. Duncan’s Multiple Range Test was used to determine significant differences (*p* ≤ 0.05) between substrates when needed.

## 3. Results

### 3.1. Substrate Lignocellulosic Content

Substrates differed significantly in terms of their C and N contents and C:N (*p* < 0.05). S5 had the highest C content, followed by S7, S1, and S4, whereas S2, S3, and S6 had the lowest C content (Table 2). Substrate S1 (100% sugar bagasse) had a significantly lower N content (0.52) than the other substrates (*p* < 0.05), suggesting that the higher N content of the other six substrates was due to the nutrient additives. The N content of S4 (2.04) was significantly higher than that of the other substrates (*p* < 0.05; Table 2). The C:N of the different substrates ranged from 20.96 (S4) to 88.44 (S1; Table 2). 

In addition, significant differences (*p* < 0.05) were observed between different substrates in terms of their cellulose, lignin, and hemicellulose concentrations (Table 2). S2 had the highest lignin content (26.64%) and S4 had the lowest (15.14%) (Table 2). S4 had the highest hemicellulose and lowest lignin content, while S6 had the highest cellulose content levels and the lowest hemicellulose content (Table 2).

### 3.2. Substrate Mineral Content

Substrates also differed significantly in terms of their mineral content (Table 2; *p* < 0.05). Mg, K, Zn, and Na levels were significantly higher (*p* < 0.05) in S4 than in the other substrates and Ca and Fe levels were significantly higher in S6 than in the other substrates (*p* < 0.05). Mg, Ca and K values were significantly lower in S1 and S7 and Fe and Na were significantly lower in S5 than in the other substrates (*p* < 0.05; Table 2).

### 3.3. Spawn Run Times, Pinhead Formation, and Fructification 

Spawn run times on different substrates differed significantly (*p* < 0.05; Table 3), indicating that the different nutrient additives added to the sugarcane bagasse substrate had different effects on the spawn. Spawn run time ranged from 26.33 (S4) to 35.66 days (S1) (Table 3). The spawn run times of S5, S6, and S7 did not differ significantly (*p* > 0.05). S1 had the longest spawn run time (33 days) followed by S5. Pinhead formation was fastest on S4 (36 days) and slowest on S7 (45 days), followed by S1, S6, and S2. However, the number of days required for pinhead formation on substrates S1, S2, S3, S5, S6, and S7 did not vary significantly (Table 3). Similarly, the number of days that elapsed between the spawn run and pinhead formation also varied significantly (*p* < 0.05; Table 3). 

Fructification on different substrates differed significantly (*p* < 0.05; Table 3). Fructification occurred after significantly fewer days on S2 than on S1 (*p* < 0.05); however, the time to fructification on S1 and S2 did not differ significantly from that on S3–S7 (*p* > 0.05; Table 3). The number of days to first harvest did not differ significantly among the substrates (*p* > 0.05; Table 3). However, the number of fruit bodies varied significantly (*p* < 0.05) among substrates (Table 3). Fruitbody formation was highest on S4 (*p* < 0.05; 15.66), followed by S2 (11.33) and S3 (11.33), and S6 (7.66). Fruitbody formation was lowest on S1, S5, and S7—production levels on these substrates did not differ significantly (*p* > 0.05). Interestingly, the sporocarps that developed on all substrates were of a marketable quality (Figure 2F), with a light- to dark-brown fleshy convex cap, creamy white gills, and a light brown, slightly tough stipe (Figure 2). 

### 3.4. Cap Diameter and Stipe Length 

Shiitake mushroom cap diameter and stipe length were significantly affected by the type of substrate (*p* < 0.05; Table 3). The cap diameter of mushrooms that developed on substrates S3 (13.33 cm) and S2 (12.33 cm) were significantly larger (*p* < 0.05) and the cap diameters of mushrooms that developed on substrates S1, S5, and S7 were significantly smaller than those that developed on other substrates (*p* < 0.05). The stipes of mushrooms that developed on S3 and S6 (8.53 cm) were significantly longer (*p* < 0.05) and those that developed on S5 (5.56 cm) were significantly shorter than those that developed on other substrates (*p* < 0.05). The stipe length of mushrooms that developed on substrate S5 did not differ significantly from those that developed on S1 and S4 (*p* < 0.05; Table 3).

### 3.5. Total Yield and Biological Efficiency 

Shiitake mushroom yield and biological efficiency were significantly affected by substrate type (*p* < 0.05; Figure 3A). Significantly higher yield (434.33 g/500 g of substrate) and biological efficiency (86.83%) were obtained using substrate S4 (*p* > 0.05) and significantly lower yield (120.33 g/500 g) and biological efficiency (24.33%) were obtained using substrate S7 than when using other substrates (*p* > 0.05).

In general, the first flush was harvested 2 to 3 months after substrates were inoculated with spawn, and high yields were obtained using all substrates except for S5, S6, and S7 (Figure 3B). Substrates S2 and S4 promoted early fructification and an early cropping period, which also resulted in significantly higher total yields (Figure 3A). With respect to yield distribution among flushes, although flushes were harvested over a 1-to-3-month cycle, the highest yields were obtained from the first flushes on S2 and S4. The highest yield of the second flushes was obtained on S6 followed by S4. By contrast, the highest yield obtained from the third flushes was obtained on S5.

## 4. Discussion

As anticipated, this study revealed that substrates enriched with cow dung or horse or chicken manure (S2, S3 and S4, respectively) had the lowest C:N and produced the highest mushroom yields. This is because mushroom production is greater on substrates that provide the optimum C:N ratio for high growth [36,37]. This is likely to be because C is an important component of the energy and structural needs of the mushroom [38] and N is needed for the synthesis of amino acids, which are building blocks for protein [16]. Abdullah et al. [14] reported that substrates with C:N ratios of 20.38 and 25.10 supported faster growth of *L. edodes* than corn cobs (C:N 47.55), indicating that the mycelium extension rate is related to the bioavailability of N. This study further supports these findings given that the highest total yields were obtained on substrate S4 (Figure 3), which had the lowest C:N (20.96) (Table 2). However, the other six substrates had a higher C:N ratio than the recommended C:N ratio for mushroom substrates, indicating that a high C:N (more C) can limit the availability of N, leading to slow growth and low mushroom yields [39]. This suggests that a substrate formulation with a low C:N is crucial for obtaining high yields of shiitake mushrooms. This can be achieved through the careful selection and mixing of substrate materials to create a balanced C:N that meets the specific requirements of the mushroom species being cultivated.

In this study, we also evaluated whether the cellulose, lignin, and hemicellulose contents of the formulated substrates affected the yield of shiitake mushrooms. Substrates with a high proportion of lignocellulosic materials generally have a low protein content and, hence, mushroom yields on this type of substrate are low. In this study, the S4 substrate, which comprised 80% sugarcane bagasse with 20% chicken manure, had a significantly lower lignin and cellulose content but a higher hemicellulose content than other substrates. A significantly higher yield of shiitake mushrooms was obtained on S4 compared with the yield on other substrates. Compared to other studies, the result of this study indicates that sugarcane bagasse contains sufficient cellulose and hemicellulose materials for the cultivation of *L. edodes* mushrooms [22,40]. This is because substrates that have a high ratio of easily digestible carbohydrates, such as hemicellulose, can support rapid growth and high yields of mushrooms [14]. This suggests that a substrate with a high proportion of hemicellulose is preferable for shiitake mushroom production [41]. Furthermore, substrate S4 supported higher yields within a shorter cropping period and had higher biological efficiency than the other substrates [22,40]. A similar enhancement in mushroom yield and a reduction in the time needed for *L. edodes* fruiting were reported when sawdust-based substrates were supplied with saccharide amendments [42]. However, excessive lignin content in substrates derived from agricultural by-products can make substrates less accessible to enzymes and, thus, these substrates may require long cultivation times to improve the accessibility of the carbohydrates for mushroom growth. Substrates with a high lignin content are therefore less preferable for mushroom growth because mushrooms take more time to develop [42]. This study also assessed the impact of different minerals on shiitake mushroom cultivation. Substrates with a balanced C:N ratio also had a high macro-nutrient (K, Mg and Ca) and micro-nutrient (Zn, Fe and Na) content. For instance, substrate S4 contained 0.32% Ca and 0.29% Mg, which were much higher values than the minimum recommended levels of 0.1% Ca and 0.05% Mg for the growth of most mushrooms, including shiitake [19]. Shiitake mushrooms require sufficient levels of K, Mg, and Ca for their healthy growth and development [43,44]. K is crucial for regulating water uptake, enhancing tolerance to environmental stress, and improving yield while Mg and Ca are necessary for the uptake and transportation of other nutrients, such as N [45]. Inadequate levels of these minerals in the substrate can impede the absorption of necessary nutrients, leading to poor growth and fruiting of shiitake mushrooms [44]. Furthermore, shiitake mushrooms prefer a slightly acidic pH range of around 5.5 to 6.5 [44]. Ca can help buffer the substrate to prevent the pH from becoming too acidic, while Mg can prevent the pH from becoming too alkaline [46]. Overall, this study highlights the importance of selecting a substrate with the correct balance of macro-elements, C:N ratio, and lignocellulolytic materials to ensure the healthy growth and development of shiitake mushrooms.

Among the different substrates, fruiting bodies that developed on S3 had the largest cap length (13.33 cm) and longest stripes (8.53 cm) (Table 3). These values are higher than those reported by Ozcelik and Peksen [47], who recorded shiitake mushrooms with a pileus diameter of 6.83 cm and a stipe length of 1.3 cm on a wheat straw substrate. However, they obtained a higher total number of fruiting bodies (34.8) compared to the present study (15.66 on S4).

Spawn run times, pinhead formation, and fructification of shiitake mushrooms were affected by the substrate. Studies have shown that the bioconversion efficiency of the substrate and successful cultivation of *L. edodes* primarily depends on the initial stage of mycelial growth and complete colonization of the substrate [14]. Achieving rapid colonization during this stage is crucial to minimize the risk of contamination during the early stages of the cultivation process. The present study suggests that the duration of the spawn run time is a crucial factor in determining the efficiency of the substrate bioconversion rate. Specifically, the combination of 80% sugar cane bagasse with 20% chicken manure led to a shorter spawn running time and earlier pinhead formation, while the 80% sugar cane bagasse with 20% cow dung resulted in fructification starting earlier than on other substrates. The N source used in the substrate preparation (i.e., chicken manure or cow dung) played a significant role in these differences. Both chicken manure and cow dung are good sources of N for mushroom cultivation due to their high levels of organic compounds such as urea, uric acid, and proteins [48,49]. This study also observed that the number of flushes, the yield obtained at each flush and the overall yield (i.e., the sum of the yields of all the flushes) varied depending on the substrate, indicating that substrate composition influences the growing conditions and, hence, sporocarp formation. For example, compared with the other substrates, the S4 substrate promoted higher total productivity over a shorter period. Although sugarcane bagasse itself is rich in nutrients, it has only small amounts of readily available N, which could explain the lower mushroom yields on S1 and on S7 (when sugarcane bagasse was used in conjunction with cotton seed hulls, which have a relatively high lignin content) compared with other substrates. However, when sugarcane bagasse was mixed with other materials with sufficient nutrients for mushroom growth in a controlled environment, high yields were obtained. Hence, the substrate formulations examined in this study exhibit promise for shiitake mushroom production. These formulations are specifically designed to foster mycelial growth and colonization, thereby maximizing the production of fruiting bodies and ultimately increasing the overall mushroom yield [13,50]. Overall, across all substrates, the initial harvest yielded higher results except for substrates S5 and S6. However, subsequent harvests in all substrates showed lower or irregular yields, suggesting that the total cultivation output of shiitake mushrooms can be impacted by factors beyond substrate type, such as spawn quality, environmental conditions, and cultivation techniques. Thus, to maximize the total mushroom yield, it is necessary to optimize the conditions for each flush. This could involve adjusting factors such as temperature, humidity, lighting, and nutrient levels, as well as harvesting and maintaining the growth medium.

## 5. Conclusions

Land scarcity and agricultural land fertility degradation contribute to food insecurity in Ethiopia, which has led to a search for alternative food sources. Edible mushrooms are seen as a potential solution to increase dietary diversity and achieve food security. Thus, this study aimed to formulate a substrate that would help to optimize mushroom production. Current findings show that sugarcane bagasse supplemented with chicken manure or cow dung provided an ideal C:N ratio for shiitake mushroom production as well as other important macro- and micronutrients. Therefore, a substrate comprising a mixture of sugar bagasse and chicken manure would be a viable option for cultivating shiitake mushrooms, particularly in regions where chicken manure is readily available. Alternatively, sugar bagasse with cow dung would be a suitable alternative as this substrate also produced high yields and faster mushroom production compared with some of the other substrates assessed in this study. However, proper preparation of the chicken manure and cow dung is crucial to avoid contamination and ensure safety.

## Figures and Tables

**Figure 1 jof-09-00811-f001:**
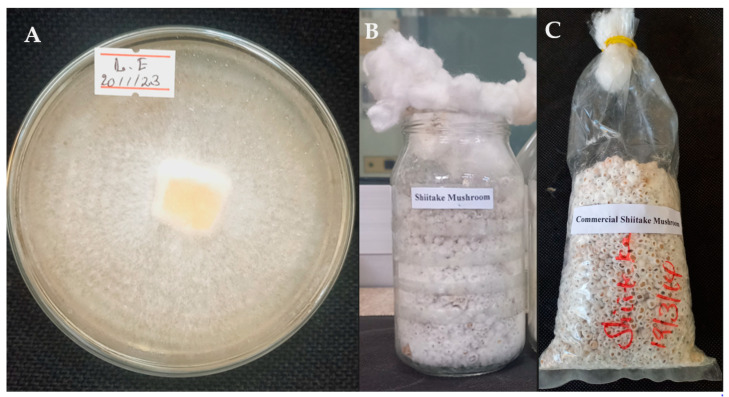
Fourteen-day-old colony of *Lentinula edodes* on potato dextrose agar (**A**), 18-day-old mother spawn (**B**) and 12-day-old commercial spawn (**C**) for growing shiitake mushrooms.

**Figure 2 jof-09-00811-f002:**
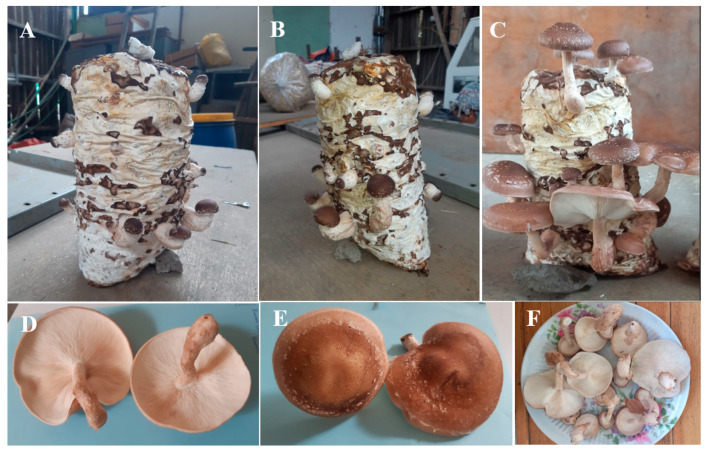
Shiitake fruiting bodies produced on a sugarcane bagasse substrate. Pinhead initiation and sporophore induction (**A**,**B**); mature fruiting bodies ready for harvest (**C**); underside of the fruit body showing the gills (**D**); cap (**E**); marketable mushrooms (**F**). The photo taken form the S4 substrate.

**Figure 3 jof-09-00811-f003:**
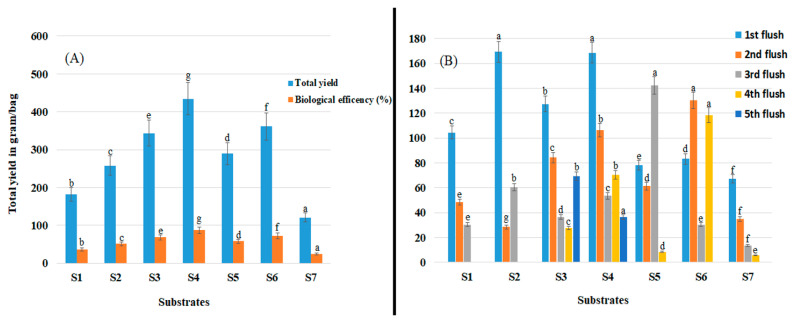
Shiitake mushroom yield on seven different substrates (S1–S7). (**A**) Total yield and biological efficiency (%). (**B**) Yield obtained from each flush and the number of flushes. Data are mean values ± the standard error of the mean. Values with the same letter are not significantly different.

**Table 1 jof-09-00811-t001:** Substrate formulations used for the cultivation of *Lentinula edodes*.

Substrate Code	Formulation
S1	100% sugarcane bagasse
S2	80% sugarcane bagasse with 20% cow dung
S3	80% sugarcane bagasse with 20% horse manure
S4	80% sugarcane bagasse with 20% chicken manure
S5	80% sugarcane bagasse with 20% cotton seed hulls
S6	80% sugarcane bagasse with 20% sugarcane filter cake
S7	80% sugarcane bagasse with 20% sugarcane trash

**Table 2 jof-09-00811-t002:** Lignocellulosic composition and mineral content of substrates used for shiitake mushroom cultivation ^1^.

Substrates ^2^	Composition (% Dry Weight)						Mineral Elements (mg kg^−1^)					
	C	N	C/N	Cellulose	Lignin	Hemicellulose	Mg	K	Ca	Zn	Fe	Na
S1	46.07 ± 0.21 ^c^	0.52 ± 0.01 ^g^	88.44 ± 1.72 ^a^	40.16 ± 0.55 ^b^	16.83 ± 0.05 ^e^	17.70 ± 0.00 ^b^	627.53 ± 0.05 ^b^	3591.20 ± 0.10 ^a^	1040.20 ± 0.10 ^a^	8.18 ± 0.00 ^b^	2010.20 ± 0.01 ^d^	1160.20 ± 0.10 ^d^
S2	40.57 ± 0.51 ^e^	1.28 ± 0.65 ^b^	31.61 ± 0.13 ^f^	30.03 ± 0.14 ^f^	26.64 ± 0.19 ^a^	12.88 ± 0.26 ^c^	2748.53 ± 0.05 ^f^	7125.53 ± 0.05 ^c^	14,107.53 ± 0.57 ^f^	34.09 ± 0.00 ^f^	1645.06 ± 0.05 ^c^	1560.30 ± 0.05 ^f^
S3	40.56 ± 0.05 ^e^	1.05 ± 0.13 ^e^	38.54 ± 0.20 ^e^	39.42 ± 0.05 ^c^	20.46 ± 0.05 ^b^	9.95 ± 0.24 ^d^	1683.75 ± 0.00 ^e^	8210.05 ± 0.05 ^f^	4230.06 ± 0.057 ^d^	19.09 ± 0.00 ^d^	2515.06 ± 0.05 ^e^	1330.03 ± 0.05 ^e^
S4	42.78 ± 0.67 ^d^	2.04 ± 0.87 ^a^	20.96 ± 0.04 ^g^	29.13 ± 0.01 ^g^	15.14 ± 0.27 ^f^	21.96 ± 6.05 ^a^	2858.75 ± 0.00 ^g^	11,408.06 ± 0.05 ^g^	3235.26 ± 0.15 ^c^	56.59 ± 0.00 ^g^	3590.06 ± 0.05 ^f^	2245.06 ± 0.06 ^g^
S5	49.48 ± 0.01 ^a^	0.61 ± 0.34 ^f^	80.72 ± 0.73 ^b^	37.45 ± 0.48 ^e^	26.64 ± 0.35 ^a^	12.88 ± 0.26 ^c^	873.73 ± 0.00 ^c^	7145.53 ± 0.05 ^d^	4940.13 ± 0.15 ^e^	12.84 ± 0.00 ^c^	1061.84 ± 0.01 ^a^	780.10 ± 0.10 ^a^
S6	40.56 ± 0.43 ^e^	0.84 ± 0.44 ^d^	48.00 ± 0.28 ^d^	40.78 ± 0.01 ^a^	19.73 ± 0.05 ^c^	13.15 ± 0.43 ^c^	1598.75 ± 0.00 ^d^	4370.53 ± 0.05 ^b^	15,857.56 ± 0.05 ^g^	29.10 ± 0.01 ^e^	4270.03 ± 0.05 ^g^	1125.06 ± 0.05 ^c^
S7	47.18 ± 0.19 ^b^	0.73 ± 0.18 ^e^	63.84 ± 0.09 ^c^	38.37 ± 0.01 ^d^	17.64 ± 0.18 ^d^	18.95 ± 0.08 ^ab^	622.75 ± 0.00 ^a^	7305.46 ± 0.05 ^e^	1660.10 ± 0.10 ^b^	6.32 ± 0.00 ^a^	1517.50 ± 0.10 ^b^	839.06 ± 0.05 ^b^

^1^ Values in parenthesis are the standard deviation. Values within a column with different superscript letters are significantly different (*p* < 0.05).^2^ S1, 100% sugarcane bagasse; S2, 80% sugarcane bagasse + 20% cow dung; S3, 80% sugarcane bagasse + 20% horse manure; S4, 80% sugarcane bagasse + 20% chicken manure; S5, 80% sugarcane bagasse + 20% cotton seed hulls; S6, 80% sugarcane bagasse + 20% sugarcane filter cake; S7, 80% sugarcane bagasse + 20% sugarcane trash.

**Table 3 jof-09-00811-t003:** Morphological parameters and characteristics of shiitake fruiting bodies on different substrates ^1^.

Substrate	Spawn Run Time (Days)	Pinhead Formation (Days)	Fructification (Days)	First Harvest (Days)	Cap Diameter (cm)	Stipe Length (cm)	Total No. Fruiting Bodies
S1	35.66 ± 057 ^a^	43.66 ± 0.57 ^ab^	5.00 ± 1.45 ^a^	4.66 ± 0.57 ^a^	4.66 ± 0.57 ^c^	6.16 ± 0.28 ^d^	5.33 ± 0.57 ^d^
S2	28.66 ± 0.57 ^c^	42.66 ± 5.85 ^b^	3.33 ± 0.57 ^b^	4.33 ± 0.57 ^a^	12.33 ± 0.57 ^a^	7.70 ± 0.17 ^b^	11.33 ± 0.57 ^b^
S3	28.00 ± 1.34 ^cd^	39.00 ± 1.24 ^a^	4.00 ± 1.00 ^ab^	4.11 ± 1.00 ^a^	13.33 ± 0.57 ^a^	8.53 ± 0.05 ^a^	11.33 ± 0.57 ^b^
S4	26.33 ± 0.57 ^d^	36.00 ± 1.52 ^c^	3.66 ± 0.57 ^ab^	3.66 ± 0.05 ^a^	10.33 ± 0.57 ^b^	6.66 ± 0.57 ^c^	15.66 ± 0.28 ^a^
S5	33.00 ± 2.64 ^b^	40.00 ± 1.34 ^bc^	4.66 ± 0.57 ^ab^	4.33 ± 0.57 ^a^	5.56 ± 0.57 ^c^	6.53 ± 0.05 ^cd^	5.00 ± 1.60 ^d^
S6	31.66 ± 0.57 ^b^	44.33 ± 2.08 ^ab^	4.00 ± 1.00 ^ab^	4.13 ± 1.02 ^a^	10.00 ± 1.04 ^b^	8.53 ± 0.05 ^a^	7.66 ± 0.57 ^c^
S7	32.33 ± 0.57 ^b^	45.00 ± 1.16 ^a^	4.66 ± 0.57 ^ab^	4.66 ± 0.57 ^a^	4.66 ± 0.28 ^c^	5.56 ± 0.05 ^e^	5.33 ± 0.57 ^d^

^1^ Values are expressed as means ± the standard deviation. Means within the same column followed by the same letter are not significantly different at *p* ≤ 0.05 according to Duncan’s multiple range test.

## Data Availability

Not applicable.

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
