# Peer review of "Substrate Optimization for Shiitake (Lentinula edodes (Berk.) Pegler) Mushroom Production in Ethiopia"

_jof, 2023, doi:10.3390/jof9080811_

Round 1

Reviewer 1 Report

This manuscript reports the results on cultivation of Lentinula edodes in different substrates based on   sugarcane bagasse. Although this work is important because may contribute to find an alternative source of food using agricultural wastes for mushroom cultivation in Ethiopa its scientific novelty  is not so high. There are several articles reporting the use of sugarcane bagasse for the cultivation of this mushroom although using different formulations.  These articles should be cited.

The manuscript is generally clearly written only the discussion is slightly convoluted and repetitive. It needs to be partially rewritten and shortened considering the low level of novelty of this work.

There are other minor points which need attention:

Line 47 From the more recent literature L. edodes is the most cultivated mushroom now (see https://doi.org/10.1002/9781119149446.ch2)

Table 2 is not clear. I suggest to use the format of table 3

Line 191 S6 has not the highest lignin content levels. As you has previously written S2 has the highest lignin content.

Table 3 in my opinion cap diameter is a not significant parameter because it mostly depends on the harvesting time. In fig. 2F clearly show how variable is the cup size in marketable mushrooms

Figure 3 the characters are too small

Discussion 

lines 271-278 are convoluted please simplify. 

Lines 328-333 it is not possible compare the data of different experiments. There are too many different variables

Lines 360-365. In your experiment spawn quality, environmental conditions and cultivation techniques should be the same.

The references are not uniformly formatted: in some titles of articles all the words are capitalized in others all the words are not capitalized

The reference list is repeated two times

Author Response

Responses to the Reviewers

#Reviewer 1

General comment

This manuscript reports the results on cultivation of Lentinula edodes in different substrates based on sugarcane bagasse. Although this work is important because may contribute to find an alternative source of food using agricultural wastes for mushroom cultivation in Ethiopia its scientific novelty is not so high. There are several articles reporting the use of sugarcane bagasse for the cultivation of this mushroom although using different formulations. These articles should be cited. The manuscript is generally clearly written only the discussion is slightly convoluted and repetitive. It needs to be partially rewritten and shortened considering the low level of novelty of this work. There are other minor points which need attention:

  • Dear reviewer, we appreciate the positive comments and concern you have towards our work. We considered and cited those references used sugarcane bagasse for cultivation of mushroom in the revised version of the manuscript. Also, by replying to your specific comments below we hope to clarify the fair doubts raised in first version of our manuscript.

Minor comments

Line 47 From the more recent literature L. edodes is the most cultivated mushroom now (see https://doi.org/10.1002/9781119149446.ch2)

  • The reference is included as suggested and the sentence rewrite and the change is highlighted in the text.

Table 2 is not clear. I suggest using the format of table 3

  • Corrected as suggested.

Line 191 S6 has not the highest lignin content levels. As you have previously written S2 has the highest lignin content.

  • Corrected

Table 3 in my opinion cap diameter is a not significant parameter because it mostly depends on the harvesting time. In fig. 2F clearly show how variable is the cup size in marketable mushrooms

  • Dear Reviewer, we included the cap diameter as it provides valuable information about the maturity and size of the mushroom. In the realm of mushroom foraging and cooking, the cap diameter is often considered when determining the edibility and culinary suitability of a mushroom. Some mushrooms are prized for their large caps, while others are preferred when they are still young and have smaller caps. The size of the cap can also affect the texture and taste of the mushroom when cooked. Therefore, knowing the cap diameter helps in selecting mushrooms that are appropriate for various culinary purposes. Off course we remain open to clarify it further,

Figure 3 the characters are too small.

  • Dear Reviewer, we have provided the figure with the good size of the letters as shown in the figure below. But due to the format used in the manuscript, the figure seems small.

Discussion

lines 271-278 are convoluted please simplify.

  • Removed the reputation form the paragraph.

Lines 328-333 it is not possible compare the data of different experiments. There are too many different variables.

  • We have clarified above why we focused on cape diameter among the other parameters, Off course we remain open to clarify it further,

Lines 360-365. In your experiment spawn quality, environmental conditions and cultivation techniques should be the same.

  • Dear reviewer, thank you for the comment here. Yes, we have used the same type of experimental setup including the spawn quality, environmental conditions, and cultivation technique for all the treatments. This is already explained in the methodology part of the manuscript.

The references are not uniformly formatted: in some titles of articles all the words are capitalized in others all the words are not capitalized.

  • All the references are checked and corrected according to the comments provided.

The reference list is repeated two times.

  • Corrected as suggested.

Summary: We appreciate the positive comments you have regarding our study. We hope that we have provided the necessary responses for your concerns to reconsider our manuscript towards an eventual acceptance for publication. Of course, we remain open to clarifying any further concern that you might have in the revised version of the manuscript.

Reviewer 2 Report

Comments to the Author:

Title: Substrate optimization for Shiitake (Lentinula edodes (Berk.) Pegler) mushroom production in Ethiopia

Overview and general recommendation:

The manuscript deals with an important topic related to the substrate optimization for Shiitake (Lentinula edodes (Berk.) Pegler) mushroom production in Ethiopia.

The manuscript technically sounds well and shows high novelty. However, it needs moderate linguistic adjustments mainly related to length and cumbersomeness of formulated sentences. Most needed adjustments are highlighted in “Minor comments” section. Also, some statements lacked reliable sources (references) that should be provided. The list of references shall be up-to-date (enclosing references from 2010 and onwards); accordingly, for references number 11, 12, 13, 14, 16, 17, 20, 29, 36, 38, 40, 42, 43, 44, 48, and 50, it is requested to substitute them by more recent references. Additionally, the first voice form of the sentence shall be avoided as it is not very appropriate scientifically and replaced by the impersonal form instead.

The Abstract part outlines clearly the problematic, aims, and methodology of the current study while reporting the main conclusions aroused. However, it should outline better the main findings of the study: authors shall focus on the biological efficiency values obtained from each substrate, the mineral and lignocellulosic compositions, growing periods, and morphological parameters. They should also outline the percentages of improvements observed in optimal substrates compared to the others. The Introduction part is well structured and aiming and underlines appropriately the whole subject under study. The aims of the study are also clear and understood. The Materials and methods part is clear, well written, and encloses most information related to the adopted methodology, and statistical analysis. Minor adjustments are only needed in these concerns. Although it shows a correct statistical representation, the Results part needs major adjustments. The scientific analysis of the findings should be well improved. Percentages of improvements should be highlighted. The Discussion part compared adequately the findings of the present study with previous ones found in literature. It also gave valuable recommendations for mushroom farmers’ best. An appropriate Conclusions was added in which authors summarized appropriately the findings of their study and suggested further related research being based on the raised assumptions.

My comments and queries for authors are detailed below in “Major comments” and “Minor comments” sections.

1.1.            Major comments:

1-      The manuscript needs moderate linguistic adjustments mainly related to length and cumbersomeness of formulated sentences. Most needed adjustments are highlighted in “Minor comments” section.

2-     The list of references shall be up-to-date (enclosing references from 2010 and onwards); accordingly, references number 11, 12, 13, 14, 16, 17, 20, 29, 36, 38, 40, 42, 43, 44, 48, and 50 shall be substituted by more recent ones. All references’ writing form shall follow the journal’s guidelines.

3-      Abstract: The Abstract part should outline better the main findings of the study: authors shall focus on the biological efficiency values obtained from each substrate, the mineral and lignocellulosic compositions, growing periods, and morphological parameters. They should also outline the percentages of improvements observed in optimal substrates compared to the others.

4-      3. Results, 3.1. Substrate lignocellulosic content: Page 5, lines 182–186 and 189–192: The scientific analysis should be improved; the percentages of variation in comparison with the control and treatments shall be highlighted when there is significant difference.

5-      3. Results, 3.2. Substrate mineral content: Page 5, lines 195–198: Same recommendation as in the previous comment.

6-      3. Results, 3.3. Spawn run time, pinhead formation, and fructification: Same recommendation as in the previous two comments.

7-      3. Results, 3.4. Cap diameter and stipe length: Same recommendation as in the previous comments.

8-      3. Results, 3.5. Total yield and biological efficiency: Same recommendation as in the previous comments.

 1.2.            Minor comments:

9-      Abstract: Page 1, lines 16–18: “We assessed… (S7)”: Kindly avoid the first voice form of the sentence and adopt the impersonal form instead.

10-  Abstract: Page 1, line 17: Kindly remove “either” and “or”.

11-  Abstract: Page 1, lines 19–20: Kindly mention the biological efficiencies rather than the biological yields.

12-  Abstract: Page 1, line 21: Kindly adjust as follow: “produced by”.

13-  Abstract: Page 1, lines 21–22: “S4… ratio”: Kindly remove this sentence.

14-  Abstract: Page 1, lines 24 and 26: Kindly adjust as follow: “Shiitake mushroom”.

15-  1. Introduction: Page 1, line 41: Reference [8] is relatively old (older than 2010; also, not very reliable); accordingly, kindly replace it by the following suitable reference: “doi:10.1088/1755-1315/1090/1/012020”.

16-  1. Introduction: Page 1, line 44: Reference [10] is relatively old (older than 2010); accordingly, kindly replace it by the following suitable reference: “doi:10.3390/agriculture12122095”.

17-  1. Introduction: Page 2, line 51: Kindly replace “to have” by “to own”.

18-  1. Introduction: Page 2, lines 57–58: Kindly adjust as follow: “its cultivation”.

19-  1. Introduction: Page 2, lines 66–67: “Other… yields”: This statement lacks reliable sources (references); accordingly, kindly provide them.

20-  1. Introduction: Page 2, lines 69–73: “However… [23]”: The sentence is long and cumbersome; accordingly, kindly reformulate in order to make it more concise, clearer and more aiming. Moreover, kindly avoid the first voice form of the sentence and adopt the impersonal form instead.

21-  1. Introduction: Page 2, lines 73–75: “Typically… [24]”: The sentence is badly written in standard English; accordingly, kindly reformulate it.

22-  1. Introduction: Page 2, lines 75–76: “We therefore… yield”: Kindly avoid the first voice form of the sentence and adopt the impersonal form instead.

23-  1. Introduction: Page 2, line 77: Kindly adjust as follow: “the objective of this study was”.

24-  2. Materials and Methods, 2.2. Culture source and spawn preparation: Page 2, line 90: Kindly mention the composition of PDA prepared.

25-  2. Materials and Methods, 2.2. Culture source and spawn preparation: Page 3, lines 96 and 109: Kindly add a space between the numbers and the unit here.

26-  2. Materials and Methods, 2.2. Culture source and spawn preparation: Page 3, lines 97 and 109: Kindly adjust as follow: “covered by mycelium”.

27-  2. Materials and Methods, 2.3. Substrate preparation: Page 3, line 115: Kindly replace “or” by a comma “,”.

28-  2. Materials and Methods, 2.3. Substrate preparation: Page 3, line 116: Kindly remove “or”.

29-  2. Materials and Methods, 2.3. Substrate preparation: Page 3, line 127: Kindly adjust the unit following the SI standards in the whole manuscript: “g” and not “gm”.

30-  2. Materials and Methods, 2.3. Substrate preparation: Page 3, line 130: Kindly adjust as follow: “covered by”.

31-  2. Materials and Methods, 2.3. Substrate preparation: Page 4, lines 135–138: “After each… fructification”: The sentence is badly written in standard English; accordingly, kindly reformulate it.

32-   2. Materials and Methods, 2.3. Substrate preparation: Page 4, line 138: Kindly adjust as follow: “for three to five flushes”.

33-  2. Materials and Methods, 2.4. Determination of L. edodes cultivation parameters: Page 4, line 148: Kindly mention that you used a sliding calliper for mushrooms physical measurements.

34-  2. Materials and Methods, 2.5. Substrate analyses: Page 4, line 152: Kindly adjust as follow: “oven-dried”.

35-  2. Materials and Methods, 2.5. Substrate analyses: Page 4, lines 166–168: “The C:N ratio… Spectroscopy”: Kindly provide a reference for such methodology adopted.

36-  2. Materials and Methods, 2.6. Statistical analysis: Pages 4–5, lines 171–173: “We also… efficiency”: Kindly avoid the first voice form of the sentence and adopt the impersonal form instead.

37-  3. Results, 3.1. Substrate lignocellulosic content: Page 5, line 187: Kindly remove “%” as ratios have no unit.

38-  3. Results, 3.3. Spawn run time, pinhead formation, and fructification: Page 6, line 212: Kindly replace “on” by “of”.

39-  3. Results, 3.3. Spawn run time, pinhead formation, and fructification 6, line 234: Kindly adjust as follow: “(Figure 2)”.

40-  3. Results, 3.5. Total yield and biological efficiency: Page 7, lines 253–254: Kindly remove “A” before “significantly”.

41-  3. Results, 3.5. Total yield and biological efficiency: Page 7, lines 254 and 256: Kindly adjust as follow: “(p < 0.05)”.

42-  3. Results, 3.5. Total yield and biological efficiency: Page 8, line 266: Kindly adjust as follow: “flushes of”.

43-  4. Discussion: Page 8, line 270: Kindly adjust as follow: “this study”.

44-  4. Discussion: Page 8, lines 280 and 283: Kindly remove “%” after the C:N ratios values.

45-  4. Discussion: Page 8, line 281: Kindly adjust as follow: “The present study”.

46-   4. Discussion: Page 8, lines 291–292: “In this study… mushrooms”: Kindly avoid the first voice form of the sentence and adopt the impersonal form instead.

47-  4. Discussion: Page 8, line 294: Kindly remove “In our study”.

48-  4. Discussion: Page 8, lines 297–298: Kindly adjust as follow: “compared to other substrates”.

49-  4. Discussion: Page 8, lines 311–313: “In this study… cultivation”: Kindly avoid the first voice form of the sentence and adopt the impersonal form instead.

50-  4. Discussion: Page 9, line 325: Kindly adjust as follow: “this study highlighted”.

51-  4. Discussion: Page 9, line 339: Kindly adjust the sentence as follow: “The present analyses suggested…”

52-  4. Discussion: Page 9, line 341: Kindly replace “+” by “with”.

53-  4. Discussion: Page 9, lines 347–350: “We also… formation”: Kindly avoid the first voice form of the sentence and adopt the impersonal form instead.

54-  4. Discussion: Page 9, lines 357–360: “Therefore… [51,52]”: The sentence is cumbersome; accordingly, kindly reformulate in order to make it clearer and more aiming.

55-  4. Discussion: Page 9, lines 365–366: “Thus… flush”: Kindly avoid the first voice form of the sentence and adopt the impersonal form instead.

56-  5. Conclusions: Page 10, lines 371–372: “Thus… production”: Same recommendation as in the previous comment.

57-  5. Conclusions: Page 10, line 373: Kindly adjust the sentence as follow: “Current findings showed that…”

The manuscript needs moderate linguistic adjustments mainly related to length and cumbersomeness of formulated sentences. Most needed adjustments are highlighted in “Minor comments” section in my attached report.

Author Response

Responses to the Reviewers

# Reviewer 2

Overview and general recommendation:

The manuscript deals with an important topic related to the substrate optimization for Shiitake (Lentinula edodes (Berk.) Pegler) mushroom production in Ethiopia.

The manuscript technically sounds well and shows high novelty. However, it needs moderate linguistic adjustments mainly related to length and cumbersomeness of formulated sentences. Most needed adjustments are highlighted in “Minor comments” section. Also, some statements lacked reliable sources (references) that should be provided. The list of references shall be up to date (enclosing references from 2010 and onwards); accordingly, for references number 11, 12, 13, 14, 16, 17, 20, 29, 36, 38, 40, 42, 43, 44, 48, and 50, it is requested to substitute them by more recent references. Additionally, the first voice form of the sentence shall be avoided as it is not very appropriate scientifically and replaced by the impersonal form instead.

The Abstract part outlines clearly the problematic, aims, and methodology of the current study while reporting the main conclusions aroused. However, it should outline better the main findings of the study: authors shall focus on the biological efficiency values obtained from each substrate, the mineral and lignocellulosic compositions, growing periods, and morphological parameters. They should also outline the percentages of improvements observed in optimal substrates compared to the others. The Introduction part is well structured and aiming and underlines appropriately the whole subject under study. The aims of the study are also clear and understood. The Materials and methods part is clear, well written, and encloses most information related to the adopted methodology, and statistical analysis. Minor adjustments are only needed in these concerns. Although it shows a correct statistical representation, the Results part needs major adjustments. The scientific analysis of the findings should be well improved. Percentages of improvements should be highlighted. The Discussion part compared adequately the findings of the present study with previous ones found in literature. It also gave valuable recommendations for mushroom farmers’ best. An appropriate Conclusions was added in which authors summarized appropriately the findings of their study and suggested further related research being based on the raised assumptions. My comments and queries for authors are detailed below in “Major comments” and “Minor comments” sections.

  • Dear reviewer, we appreciate the positive comments you have towards our work. We considered all the general overview and recommendation given. Also, by replying to your specific comments below we hope to clarify the fair doubts raised in the first version of our manuscript.

Major comments:

The manuscript needs moderate linguistic adjustments mainly related to length and cumbersomeness of formulated sentences. Most needed adjustments are highlighted in “Minor comments” section.

  • Dear Reviewer, thank you, we have considered your comments in the revised version of the manuscript.

The list of references shall be up to date (enclosing references from 2010 and onwards); accordingly, references number 11, 12, 13, 14, 16, 17, 20, 29, 36, 38, 40, 42, 43, 44, 48, and 50 shall be substituted by more recent ones. All references’ writing forms shall follow the journal’s guidelines.

  • Dear Reviewer, we have substituted the listed references with the updated one.

Abstract: The Abstract part should outline better the main findings of the study: authors shall focus on the biological efficiency values obtained from each substrate, the mineral and lignocellulosic compositions, growing periods, and morphological parameters. They should also outline the percentages of improvements observed in optimal substrates compared to the others.

  • Dear Reviewer, in the abstract section, we have emphasized the primary findings of the study due to the constraint of limited word count allotted for the abstract according to the journal’s guidelines. However, based on your suggestion we have included the biological efficiency of the mushroom with respect to the substrate formulations.

Results, 3.1. Substrate lignocellulosic content: Page 5, lines 182–186 and 189–192: The scientific analysis should be improved; the percentages of variation in comparison with the control and treatments shall be highlighted when there is significant difference.

  • Corrected as suggested.

Results, 3.2. Substrate mineral content: Page 5, lines 195–198: Same recommendation as in the previous comment.

  • Corrected as suggested.

Results, 3.3. Spawn run time, pinhead formation, and fructification: Same recommendation as in the previous two comments.

  • Corrected as suggested.

Results, 3.4. Cap diameter and stipe length: Same recommendation as in the previous comments.

  •  

Results, 3.5. Total yield and biological efficiency: Same recommendation as in the previous comments.

  •  

 Minor comments:

Abstract: Page 1, lines 16–18: “We assumed… (S7)”: Kindly avoid the first voice form of the sentence and adopt the impersonal form instead.

  • Corrected as suggested.

Abstract: Page 1, line 17: Kindly remove “either” and “or”.

  •  

Abstract: Page 1, lines 19–20: Kindly mention the biological efficiencies rather than the biological yields.

  • Corrected as suggested.

Abstract: Page 1, line 21: Kindly adjust as follow: “produced by”.

  • Corrected as suggested.

Abstract: Page 1, lines 21–22: “S4… ratio”: Kindly remove this sentence.

  •  

Abstract: Page 1, lines 24 and 26: Kindly adjust as follow: “Shiitake mushroom”.

  • Corrected as suggested.

Introduction: Page 1, line 41: Reference [8] is relatively old (older than 2010; also, not very reliable); accordingly, kindly replace it by the following suitable reference: “doi:10.1088/1755-1315/1090/1/012020”.

  • Dear Reviewer, thank you for the suggestion, the reference is now included in the text of the revised version of the manuscript.

Introduction: Page 1, line 44: Reference [10] is relatively old (older than 2010); accordingly, kindly replace it by the following suitable reference: “doi:10.3390/agriculture12122095”.

  • Included as suggested.

Introduction: Page 2, line 51: Kindly replace “to have” by “to own”.

  • Changed as suggested.

Introduction: Page 2, lines 57–58: Kindly adjust as follow: “its cultivation”.

  • Corrected as suggested.

Introduction: Page 2, lines 66–67: “Other… yields”: This statement lacks reliable sources (references); accordingly, kindly provide them.

  • The reference is provided as suggested.

Introduction: Page 2, lines 69–73: “However… [23]”: The sentence is long and cumbersome; accordingly, kindly reformulate in order to make it more concise, clearer and more aiming. Moreover, kindly avoid the first voice form of the sentence and adopt the impersonal form instead.

  • We have modified the sentence as suggested.

Introduction: Page 2, lines 73–75: “Typically… [24]”: The sentence is badly written in standard English; accordingly, kindly reformulate it.

  • Corrected as suggested.

Introduction: Page 2, lines 75–76: “We therefore… yield”: Kindly avoid the first voice form of the sentence and adopt the impersonal form instead.

  • Corrected as suggested.

Introduction: Page 2, line 77: Kindly adjust as follow: “the objective of this study was”.

  • Corrected as suggested.

Materials and Methods, 2.2. Culture source and spawn preparation: Page 2, line 90: Kindly mention the composition of PDA prepared.

  • The detail of the PDA is provided as suggested.

Materials and Methods, 2.2. Culture source and spawn preparation: Page 3, lines 96 and 109: Kindly add a space between the numbers and the unit here.

  • Corrected as suggested.

Materials and Methods, 2.2. Culture source and spawn preparation: Page 3, lines 97 and 109: Kindly adjust as follow: “covered by mycelium”.

  • Corrected as suggested.

Materials and Methods, 2.3. Substrate preparation: Page 3, line 115: Kindly replace “or” by a comma “,”.

  • Corrected as suggested.

Materials and Methods, 2.3. Substrate preparation: Page 3, line 116: Kindly remove “or”.

  • Corrected as suggested.

Materials and Methods, 2.3. Substrate preparation: Page 3, line 127: Kindly adjust the unit following the SI standards in the whole manuscript: “g” and not “gm”.

  • Corrected as suggested.

Materials and Methods, 2.3. Substrate preparation: Page 3, line 130: Kindly adjust as follow: “covered by”.

  • Corrected as suggested.

Materials and Methods, 2.3. Substrate preparation: Page 4, lines 135–138: “After each… fructification”: The sentence is badly written in standard English; accordingly, kindly reformulate it.

  • Corrected as suggested.

Materials and Methods, 2.3. Substrate preparation: Page 4, line 138: Kindly adjust as follow: “for three to five flushes”.

  • Corrected as suggested.

Materials and Methods, 2.4. Determination of L. edodes cultivation parameters: Page 4, line 148: Kindly mention that you used a sliding calliper for mushrooms physical measurements.

  • Included as suggested.

Materials and Methods, 2.5. Substrate analyses: Page 4, line 152: Kindly adjust as follow: “oven-dried”.

  • Corrected as suggested.

Materials and Methods, 2.5. Substrate analyses: Page 4, lines 166–168: “The C:N ratio… Spectroscopy”: Kindly provide a reference for such methodology adopted.

  • Provided as suggested.

Materials and Methods, 2.6. Statistical analysis: Pages 4–5, lines 171–173: “We also… efficiency”: Kindly avoid the first voice form of the sentence and adopt the impersonal form instead.

  • Removed and corrected.

Results, 3.1. Substrate lignocellulosic content: Page 5, line 187: Kindly remove “%” as ratios have no unit.

  • Removed as suggested.

Results, 3.3. Spawn run time, pinhead formation, and fructification: Page 6, line 212: Kindly replace “on” by “of”.

  •  

Results, 3.3. Spawn run time, pinhead formation, and fructification 6, line 234: Kindly adjust as follow: “(Figure 2)”.

  • Corrected accordingly.

Results, 3.5. Total yield and biological efficiency: Page 7, lines 253–254: Kindly remove “A” before “significantly”.

  • Removed as suggested.

Results, 3.5. Total yield and biological efficiency: Page 7, lines 254 and 256: Kindly adjust as follow: “(p < 0.05)”.

  • Adjusted as suggested.

Results, 3.5. Total yield and biological efficiency: Page 8, line 266: Kindly adjust as follow: “flushes of”.

  • Corrected as suggested.

Discussion: Page 8, line 270: Kindly adjust as follow: “this study”.

  • Corrected as suggested.

Discussion: Page 8, lines 280 and 283: Kindly remove “%” after the C:N ratios values.

  • Removed as suggested.

Discussion: Page 8, line 281: Kindly adjust as follow: “The present study”.

  • Corrected as suggested.

Discussion: Page 8, lines 291–292: “In this study… mushrooms”: Kindly avoid the first voice form of the sentence and adopt the impersonal form instead.

  • Corrected as suggested.

Discussion: Page 8, line 294: Kindly remove “In our study”.

  • Corrected as suggested.

Discussion: Page 8, lines 297–298: Kindly adjust as follow: “compared to other substrates”.

  • Corrected as suggested.

Discussion: Page 8, lines 311–313: “In this study… cultivation”: Kindly avoid the first voice form of the sentence and adopt the impersonal form instead.

  • Corrected as suggested.

Discussion: Page 9, line 325: Kindly adjust as follow: “this study highlighted”.

  •  

Discussion: Page 9, line 339: Kindly adjust the sentence as follow: “The present analyses suggested…”

  • Corrected as suggested.

Discussion: Page 9, line 341: Kindly replace “+” by “with”.

  • Replaces as suggested.

Discussion: Page 9, lines 347–350: “We also… formation”: Kindly avoid the first voice form of the sentence and adopt the impersonal form instead.

  • Changed as suggested.

Discussion: Page 9, lines 357–360: “Therefore… [51,52]”: The sentence is cumbersome; accordingly, kindly reformulate to make it clearer and more aiming.

  • Changed as suggested.

Discussion: Page 9, lines 365–366: “Thus… flush”: Kindly avoid the first voice form of the sentence and adopt the impersonal form instead.

  • Corrected as suggested.

Conclusions: Page 10, lines 371–372: “Thus… production”: Same recommendation as in the previous comment.

  • Corrected as suggested.

Conclusions: Page 10, line 373: Kindly adjust the sentence as follow: “Current findings showed that…”

  • Corrected as suggested.

Summary: We appreciate the positive comments you have regarding our study. We hope that we have provided the necessary responses for your concerns to reconsider our manuscript towards an eventual acceptance for publication. Of course, we remain open to clarifying any further concern that you might have.

Round 2

Reviewer 1 Report

Dear Authors the manuscript is improved but you do not make all the requested changes:

-table 2 was not modified

-You did not add any  citations and the comments of other articles using sugarcane bagasse

-you have not uniformed all the references

-Lines 360-365. These lines are still not clear It seems that in your experiment spawn quality, environmental conditions and cultivation techniques are not  the same. Please change the sentence

Author Response

Responses to the Reviewers

#Reviewer 1

General comment

Dear Authors, the manuscript is improved but you do not make all the requested changes:

  • Dear reviewer, we appreciate the positive comments you have towards our work. Also, by replying to your specific comments below we hope to clarify the fair doubts raised in first version of our manuscript.

Table 2 was not modified.

  • The table is now modified similar with that of the Table 3 as suggested previously.

You did not add any citations and the comments of other articles using sugarcane bagasse.

  • Dear Reviewer, we have included the following three articles related to Bagas as a substrate for mushroom production.

Ahmad Zakil, F.; Muhammad Hassan, K.H.; Mohd Sueb, M.S.; Isha, R. Growth and yield of Pleurotus ostreatus using sugarcane bagasse as an alternative substrate in Malaysia. IOP Conf. Ser. Mater. Sci. Eng. 2020, 736, 022021, doi:10.1088/1757-899X/736/2/022021.

Sidana, A.; Farooq, U. Sugarcane bagasse : a potential medium for fungal cultures. Chinese J. Biol. 2014, 5, 1–5.

Kumla, J.; Suwannarach, N.; Sujarit, K.; Penkhrue, W.; Kakumyan, P.; Jatuwong, K.; Vadthanarat, S.; Lumyong, S. Cultivation of mushrooms and their lignocellulolytic enzyme production through the utilization of agro-industrial waste. Molecules 2020, 25, 2811, doi:10.3390/molecules25122811.

You have not uniformed all the references.

  • Dear Reviewer, we have assessed all the references and corrected uniformly following your comment.

Lines 360-365. These lines are still not clear It seems that in your experiment spawn quality, environmental conditions and cultivation techniques are not the same.

  • Dear Reviewer, we have rephrased the sentence and is now clearer for the reader.

Summary: We appreciate the positive comments you have regarding our study. We hope that we have provided the necessary responses for your concerns to reconsider our manuscript towards an eventual acceptance for publication. Of course, we remain open to clarifying any further concern that you might have.

Reviewer 2 Report

Comments to the Author:

Title: Substrate optimization for Shiitake (Lentinula edodes (Berk.) Pegler) mushroom production in Ethiopia

Overview and general recommendation:

Authors have made all needed improvements to their manuscript and are well thanked for that. I have no more comments/suggestions to add.

Therefore, based on the overall evaluation of the manuscript, I find it suitable for publication in current form.

Very minor linguistic mistakes are detected; they can be adjusted during pre-proof stage so no need for further adjustments by the authors.

Author Response

We appreciate the positive comments you have regarding our study. We hope that we have provided the necessary responses for your concerns. Of course, we remain open to correct the possible languaje mistakes during MS Proofs as you suggested.